# Dietary Regulation of Oxidative Stress in Chronic Metabolic Diseases

**DOI:** 10.3390/foods10081854

**Published:** 2021-08-11

**Authors:** Shuai Jiang, Hui Liu, Chunbao Li

**Affiliations:** Key Laboratory of Meat Processing and Quality Control, MOE, Key Laboratory of Meat Processing, MARA, Jiangsu Innovative Center of Meat Production, Processing and Quality Control, College of Food Science and Technology, Nanjing Agricultural University, Nanjing 210095, China; 2018208011@njau.edu.cn (S.J.); 2019208023@njau.edu.cn (H.L.)

**Keywords:** oxidative stress, dietary mode, obesity, chronic metabolic inflammation, neurogenerative diseases

## Abstract

Oxidative stress is a status of imbalance between oxidants and antioxidants, resulting in molecular damage and interruption of redox signaling in an organism. Indeed, oxidative stress has been associated with many metabolic disorders due to unhealthy dietary patterns and may be alleviated by properly increasing the intake of antioxidants. Thus, it is quite important to adopt a healthy dietary mode to regulate oxidative stress and maintain cell and tissue homeostasis, preventing inflammation and chronic metabolic diseases. This review focuses on the links between dietary nutrients and health, summarizing the role of oxidative stress in ‘unhealthy’ metabolic pathway activities in individuals and how oxidative stress is further regulated by balanced diets.

## 1. Introduction

Oxidative stress is defined as a status of imbalance between oxidants and antioxidants in favor of oxidants and leads to the interruption of redox signals. It is related to chronic inflammation and the development of metabolic diseases [1]. Reactive oxygen species (ROS) may induce protein and lipid oxidation, leading to the impairment of energy metabolism, cell signaling and cell cycle, and nutrient transport and to dysfunction of biological activities [2,3].

Diets, especially high-fat or high-carbohydrate diets, have been shown to be associated with oxidative stress by elevating the levels of protein carbonylation and lipid peroxidation products while reducing the antioxidant defense status [1]. In obesity, chronic oxidative stress and associated inflammation are the underlying factors that cause insulin resistance, metabolic dysfunction, diabetes, and cardiovascular disease by disrupting signaling and metabolism [4,5]. Obesity-associated insulin resistance greatly increases oxidative stress and the risk of hypertension, dyslipidemia, type 2 diabetes, atherosclerosis, and nonalcoholic fatty liver disease [6].

On the other hand, healthy diets play a crucial role in maintaining cell and tissue homeostasis, inhibiting inflammation, and preventing chronic metabolic diseases [7]. Diet intervention has been recommended to alleviate or prevent some metabolic disorders [8]. Different foods and dietary patterns may show different impacts on health because of the diversity of diet composition. Animal-based foods have high levels of protein, fat, minerals, and B vitamins, while plant-based foods contain more carbohydrates and phytochemicals (e.g., polyphenols). In recent years, the Mediterranean diet has been widely recommended as a healthy one because it comprises mainly plant-based foods rich in antioxidants [9]. These compounds are mainly redox-derived active substances, which stimulate positive and beneficial reactions from cells by activating mild oxidative stress. Most plant polyphenols are used as antioxidants, and their roles in cell signal transduction, metabolism, and survival functions have been reported [10]. There are few reports on the relationship between dietary components and patterns characterizing the occurrence and development of diseases.

A reference search was conducted in PubMed/PMC, ScienceDirect, and Web of Science databases by inputting such keywords as “diet”, “Mediterranean diet”, “Western diet”, “oxidative stress”, “obesity”, “chronic metabolic disease”, “neurodegenerative disease”, “mitochondrial dysfunctions”, “wine”, and “polyphenols”, from 2000 to 2021.

In this paper, we offer a review on the definition of oxidative stress and its impact on health and on diet interventions to control metabolic diseases.

## 2. Oxidative Stress and Body Health

### 2.1. Oxidative Stress and ROS

ROS and reactive nitrogen species (RNS) are a class of active substances in aerobic organisms. ROS can cause oxidative stress, and RNS can cause nitrosative stress [11]. The formation and reactions of ROS and RNS are shown in Figure 1. The NADPH oxidase family (NOX) can generate ROS to remove pathogenic microorganisms. Nitric oxide synthase (NOS) catalyzes the breakdown of L-arginine to produce NO·in the body. The main function of superoxide dismutase (SOD) is to remove intracellular O_2_·^−^ and generate non-toxic O_2_ and mildly toxic H_2_O_2_. Peroxiredoxin (PRX) is a class of peroxidases that can regulate the transmission of reactive oxygen species and related signals in cells. Catalase (CAT) mainly removes hydrogen peroxide. Glutathione peroxidase (GPX) is an important peroxidase. The Fenton reaction is the reaction of H_2_O_2_ and divalent iron ions (Fe^2+^) to form hydroxide radicals (·OH).

The imbalance of ROS and RNS production leads to oxidative stress, which stimulates antioxidants defenses [11]. This imbalance status may lead to oxidative damage by oxidative modification of cellular macromolecules, structural tissue damage, and cell death via apoptosis or necrosis [12]. ROS are highly reactive molecules with an unpaired electron that affect biological processes in multiple ways. They include hydroxyl radicals, singlet oxygen, peroxides, and superoxide and are highly active and toxic to cells [13].

Oxidative stress damages the structure and function of proteins, lipids, and nucleic acids. ROS may induce protein oxidation and carbonylation, causing the dysfunction of the active sites of the enzymes, which further inhibits the accurate binding of substrates [14,15]. Protein damage may be associated with carbonylation, thiol oxidation, fragmentation, side-chain oxidation, unfolding and misfolding, and loss of bioactivities [16]. Lipids, especially unsaturated fatty acids, may be oxidized into malondialdehyde (MDA), 4-hydroxynonenal (4-HNE,) and isoprostanes [17]. MDA can react with cellular macromolecules such as proteins, RNA, and DNA [18]. Particularly, it reacts with DNA to form mutagenic adducts of deoxyguanosine and deoxyadenosine, such as 8-hydroxydeoxyguanosine, which is a significant index of DNA damage [18]. The molecule 4-HNE has been shown to be involved in signal transduction and affect cell cycle events [19]. Isoprostanes are a kind of potent vasoconstrictors that may induce endothelin release and proliferation of vascular smooth muscle cells [20].

In some cases, oxidative stress is mainly originated from the mitochondria because intracellular ROS are produced by oxidative phosphorylation [21]. At the same time, mitochondrial proteins, nucleic acids (e.g., mtDNA), and lipids are the targets of ROS, which are responsible for mitochondrial dysfunction [15]. H_2_O_2_, a critical ROS in cells, is mainly generated by the NADPH oxidase family (NOX) and oxidative phosphorylation in mitochondria [22,23]. H_2_O_2_ acts as a second messenger to regulate the target protein thiol switch and involves a variety of signal transduction pathways to regulate physiological and pathological processes [24]. H_2_O_2_ can be detoxificated by catalase and a variety of peroxidases [25], the ferriheme-containing enzyme participating in converting hydrogen peroxide to water [26].

As mentioned above, appropriate ROS are necessary to keep a healthy status in an organism. However excess ROS have been associated with many health problems including obesity, neurodegenerative diseases, cardiovascular diseases, and cancer [18,27].

### 2.2. Oxidative Stress and Obesity

Obesity is a chronic disease characterized by an increase in the accumulation of white adipose tissue, which is induced by diet or genetics factors. The intake of high-fat or high-carbohydrate diets may alter oxygen metabolism [28]. Lipid deposits are accompanied by ROS production. During this process, lipid peroxidation occurs to produce ROS, and if ROS production exceeds the antioxidant capacity of a cell, atherosclerosis may develop [29]. In addition, the adipose tissue produces several adipokines, e.g., proinflammatory cytokines including tumor necrosis factor alpha (TNF-α), interleukin-6 (IL-6), and leptin. IL-6 is involved in insulin resistance and glucose intolerance through negative regulation of visfatin. TNF-α also involves the development of insulin resistance and the inflammatory response [30]. Leptin improves insulin sensitivity and inhibits lipogenesis by stimulating dopamine uptake and regulating food intake to control weight [31]. Additionally, TNF-α and IL6 induce the production of ROS and oxidative stress. Thus, obesity is associated with a higher level of oxidative stress [32]. Obesity induces oxidative stress by many biochemical mechanisms, including glyceraldehyde auto-oxidation, superoxide generation from NADPH oxidases, oxidative phosphorylation, and polyol and hexosamine pathways. Besides, other factors including low antioxidant defense, chronic inflammation, hyperleptinemia, and post-meal ROS production lead to obesity by oxidative stress [28].

### 2.3. Oxidative Stress and Neurodegenerative Diseases

Neurodegenerative disorders such as Parkinson’s disease and Alzheimer’s disease occur mainly in the elderly and are characterized by progressive loss of neuron cells and impaired movement or cognitive function. One of the main characteristics of these diseases is mitochondrial dysfunction [32]. In neurons, the mitochondria play an essential role in the production of ATP to meet the energy demands for cellular processes, particularly in neurotransmitter synthesis and synaptic plasticity. Mitochondrial dysfunction is accompanied by increased mitochondrial permeability, mitochondrial disorganization, oxidative damage to mtDNA, compromised antioxidant systems, and telomere shortening [33]. Neurons are vulnerable to oxidative stress due to their high energy requirement, high content of fatty acids, high number of mitochondria, weak antioxidant defense, and low bioavailability of antioxidant molecules and antioxidant therapies [34,35].

In Alzheimer’s disease, protein aggregation (Aβ, in particular) causes the release of Ca^2+^ from the endoplasmic reticulum to the cytoplasm, which further leads to a decrease of endogenous GSH levels and the accumulation of ROS [36]. Neuronal functions are impaired, which further leads to neuroinflammation and neuronal loss [37,38].

In Parkinson’s disease, the progression of neurodegenerative synucleinopathies is associated with nitrosative stress [39]. Lipid peroxidation might induce α-synuclein aggregation [40]. Glycation may involve chemical cross-linking, proteolytic resistance, and aggregation [41].

### 2.4. Oxidative Stress and Immune Inflammation

Oxidative stress may result in compromised immune and inflammatory reactions [42]. Inflammation, linked to some diseases such as viral and microbial infections, is a natural defense against pathogens [43], exposure to toxic chemicals and radiation, chronic and autoimmune diseases, allergens, and a high-calorie diet [44]. Oxidative stress can lead to the differential expression of genes related to inflammatory pathways, possibly by activating transcription factors [45]. In fact, many chronic diseases are caused by inflammatory processes that trigger oxidative stress. Various inflammatory stimuli initiate the inflammatory process, leading to the secretion and synthesis of proinflammatory cytokines, for example, the activation of nuclear factor-kappa B/active protein-1 (NF-κB/AP-1) and the production of TNF-α, which play a key role in the inflammatory process [19].

Low glutathione levels or excess accumulation of free radicals increase the risk of chronic diseases and autoimmune diseases. Long-term oxidative stress enhances inflammation and thus stimulates apoptosis; therefore, it plays an important role in the pathogenesis of autoimmune diseases [46]. In the process of activation, the energy demands of immune cells is quite high, which requires a strict regulation of metabolic pathways. Therefore, metabolic pathways have considerable importance in the differentiation of immune cells [47].

ROS are byproducts of normal metabolism. In an organism, cells create a cascade of free radicals that cause oxidative stress. When our immune system fights against bacteria and generates inflammation, we suffer from increased oxidative stress [48]. When the body suffers from oxidative stress for a long time, it will increase the accumulation of free radicals and reduce the level of glutathione. Lower GSH levels will result in an increase in ROS, the impairment of immune responses, inflammation, and a higher susceptibility to infection [43]; as a result, the self-slowing down of the carbohydrate absorption repair system of immune system will be impaired. GSH participates in immune redox regulation by changing the disulfides between protein cysteine and glutathione [49].

## 3. Diet and Oxidative Stress

### 3.1. Dietary Pattern in Healthy People and Oxidative Stress

#### 3.1.1. Western Diet

Western diets are characterized by excess consumption of saturated fats, over-refined sugars, and animal-based protein and low consumption of plant-based fiber. People who regularly eat Western diets have been shown to have higher levels of oxidative stress and a greater risk of chronic disease.

Western diets may lead to metabolic disorders which are triggered by systemic and chronic inflammation and mainly include cardiovascular diseases, hypertension, insulin resistance, T2DM, metabolic syndrome, gout, obesity and cognitive impairment [50,51,52,53,54,55,56]. The incidence of some chronic inflammatory diseases in populations consuming Western diets is shown in Appendix A. High-fat diets induce the development of metabolic syndrome, with a high incidence of oxidative stress, atherogenic dyslipidemia, a pro-inflammatory and pro-thrombotic state, high blood pressure, central obesity, and cardiovascular disease [29]. Prashant et al. [57] found that a high-fat and low-carbohydrate diet can aggravate myocardial ischemia injury in obese rats, which is related to the increase of oxidative stress and the decrease of mitochondrial biosynthesis and antioxidant gene transcription.

Western-style foods may trigger the activation of NLRP3 inflammatory response, which causes long-term systemic inflammation. It involves the transcription and recombination of myeloid progenitor cells in the bone marrow and innate immune training [58]. In mice, functional loss of ten-eleven translocation (TET2) exacerbates atherosclerosis by increasing systemic NLRP3-dependent inflammation [59]. Most likely, western-style diet-induced disorders can cause immune–metabolic dysbiosis and systemic metabolic changes, which are reflected in the epigenetic and transcriptional modifications of somatic DNA and hematopoietic stem cells [60].

A high fat diet produces free fatty acids by increasing chylomicrons in the gut, which are absorbed by the liver. Indeed, free fatty acids can enter the mitochondria for β-oxidation or esterification to triglyceride Schiff bases [61]. Triglycerides accumulate as small droplets in hepatocytes or produce very-low-density lipoproteins, which are then transformed into low-density lipoproteins (LDL) [61]. An excessive LDL burden may form oxidized LDL (Ox-LDL) in the blood, which in turn is engulfed by macrophages and then turns into foam cells. Ultimately, foam cells form plaques through the aggregation of arterial endothelial cells, leading to cardiovascular and circulatory diseases, increasing the permeability of the blood–brain barrier [62].

Excess fat accumulation stimulates mitochondrial β-oxidation of free fatty acids, which leads to excessive use of cytochrome c oxidase electron flow, thus increasing the accumulation of ROS in cells [63]. Mitochondria are essential sources of ROS in cells, which oxidize unsaturated lipids deposited in fat and cause lipid peroxidation. ROS induced by high-fat diets may trigger pro-inflammatory signals and NF-κB transcription factors, which leads to the activation of NF-κB-dependent pro-inflammatory molecules [64]. Furthermore, excess production of nitric oxide (NO) can also lead to the accumulation of RNS by activating iNOS [65].

Excessive oxidative stress may cause cellular dysfunctions, cell death, apoptosis, cytotoxicity, and DNA mutation. Through damage of molecules and cells and dysfunction of tissues and organs, it promotes the occurrence and development of diseases in severe cases.

Taken together, high-calorie diets (high protein, carbohydrate, and/or fat) not only affect the level of oxidative stress, but also have an adverse impact on health because of the excess of free radicals (Figure 2). The imbalance of redox systems causes an increase in oxidative stress and inflammation, which affects the molecular and cellular level of aging markers, leading to a decline in the function of different tissues and organs. This can then increase the risk of chronic and debilitating diseases as well as of other diseases. Antioxidants can neutralize ROS and play an important role in preventing cell oxidative damage induced by free radicals [66]. Therapeutic and lifestyle interventions establish a bidirectional relationship between diet and health.

#### 3.1.2. Mediterranean Diet

The Mediterranean diet is characterized by plant-based foods; it is rich in vegetables, unrefined cereals, fruits, legumes, fish, olive oil, and nuts, and is characterized by a moderate intake of wine and dairy products and a low intake of meat and meat products [67]. A large number of studies have shown that the Mediterranean diet reduces weight and helps prevent type 2 diabetes, strokes, and cardiovascular disease, reducing the risk of cognitive impairment and Alzheimer’s disease and improving metabolic syndrome and other chronic diseases [68,69,70,71,72,73,74,75,76]. The statistics data on populations consuming the Mediterranean diet and the incidence of some chronic inflammatory diseases are shown in Appendix A.

Fibers present in foods reduce the glycemic index by slowing down carbohydrate absorption [77]. Dietary fibers are fermented into SCFAs that have a great impact on the host. SCFAs can also enter the circulation, thus affecting tissues, organs, and the brain by participating in metabolic processes [78]. In addition to their beneficial effects on energy and metabolism balance, SCFAs regulate immune responses and inflammatory reactions through several mechanisms [79]. Dietary interventions with low-fat diets and vitamins A and D play an effective role in reducing experimental autoimmune encephalomyelitis (EAE) symptoms by inhibiting the responsiveness of T and B cell [80].

Omega-3 fatty acids in fish, nuts, or seed oil are anti-inflammatory by binding to G-protein coupled receptor 120 (GPR120), which activates macrophages and inhibits toll-like receptor 2 (TLR2), toll-like receptor 4 (TLR4), and TNF-α signaling [81]. In addition, both docosahexaenoic acid (DHA) and eicosapentaenoic acid (EPA) are effective anti-inflammatory molecules. They are also substrates for the biosynthesis of protectin, resolvin, and maresin, all of which play a key role in the termination of inflammation [82].

Nuts and virgin olive oil in the Mediterranean diets reduces cardiac events. Polyphenols in the Mediterranean diet, which are excellent antioxidants, may also promote this effect by reducing inflammatory biomarkers, blood pressure, and the risk of type 2 diabetes and obesity [83]. They may reduce oxidative stress accumulation, which damages pancreatic β-cells that produce insulin [84]. In addition, moderate alcohol drinking can improve insulin sensitivity, reduce insulin secretion, and decrease fasting glucagon concentration [85].

#### 3.1.3. Oral Dietary Control of Oxidative Stress

Lactoferrin (LF) is the first line of defense proteins present in mammals, can treat diseases that impair immune homeostasis through immunomodulation, and has the ability to bridge innate immunity and adaptive immunity [86]. In rats, the comparison of differentially expressed genes after 6 h of LF oral and intravenous injection showed that 72.8% of gene changes after oral recombinant human LF (rhLF) administration were the same as those after the intravenous treatment. These two pathways show similarities in the upregulation of specific genes related to oxidative stress and inflammation. Therefore, LF can be supplemented through oral administration [86].

The pharmacokinetics of oral LF indicate that there are almost no immune epitopes of LF in the body’s circulation [87]. Oral LF may interact with specific receptors on intestinal epithelial cells, thereby triggering a signaling pathway that spreads systemic effects [88]. In addition, because LF has the ability to chelate free iron, it is known that LF can prevent damage-induced oxidative stress that leads to severe necrosis in the damaged tissue. LF may prevent and treat COVID-19 infection by inhibiting the “cytokine storm” [89]. The preventive and therapeutic effects of LF aim to protect the integrity of the intestinal mucus layer and the composition of the microbial community. In in vitro conditions, LF has been shown to protect Caco-2 cells from intracellular oxidative stress [90]. Since imbalances in the intestinal microbiota may also increase the pathogenicity of COVID-19 infection, the beneficial effect of LF on the composition of the intestinal microbiota is crucial [91]. Therefore, nutrition containing LF can be used as a preventive measure for immune system function.

### 3.2. Diet for Elderly People and Oxidative Stress

Aging is considered a pathophysiological phenomenon related to a disorder of homeostasis, including oxidative stress [92]. Frailty of the elderly is related to the increase in biomarkers of oxidative stress and the decrease in antioxidant parameters. Too many free radicals can induce oxidative stress, leading to damage and dysfunction of important organs. Oxidative stress is also related to normal aging. The development of multiple-organ failure in the elderly is associated with the excess production of free radicals in various organs caused by oxidative stress [92]. In past decades, healthy dietary patterns, evaluated by the healthy eating index and the Mediterranean diet, have been shown to be beneficial to the health of the elderly [93]. Matsuyama et al. [94] observed that the Japanese diet is linked to a significantly lower risk of functional disability in the elderly.

Mitochondrial dysfunction induced by oxidative stress is one of the main causes of disorders of skeletal muscle and other tissues and organs [95]. Aging skeletal muscle with impaired mitochondrial biogenesis and function in the elderly causes chronic fatigue, sarcopenia, and physical hypofunction [29]. Cysteine is the precursor of glutathione synthesis [96], and cysteine supplementation can restore the reduced glutathione content in erythrocytes and reverse the increase in plasma oxidative stress markers in the elderly [97].

Antioxidants play a critical role in protecting cells from oxidative damage induced by free radicals [66]. Diets comprising nuts, fruits, and vegetables are rich in unsaturated fatty acids and polyphenols reducing lipid or protein peroxidation [31,95].

### 3.3. Diet for Athletes and Oxidative Stress

The main purpose of paying attention to athletes’ diet is to ensure that their increased energy consumption and nutritional needs are met, so to maximize the adaptation to the body loads via regulating oxidative stress [98]. If the diet does not fully meet the requirements of highly trained endurance athletes, it will not ensure maximum adaptation to highly intense and/or prolonged physical loads [99]. Therefore, an appropriate selection of food, fluids, and supplements can altogether affect an athlete’s performance. Proper dietary pattern and physical exercise are quite important for health maintenance by regulating oxidative stress (Figure 3).

In recent years, the amount of carbohydrate an athlete should ingest has become a hot topic of discussion [99]. Carbohydrates are an important energy source for cells throughout the body, especially in the brain [100]. When the substrate ATP is produced by anaerobic lactic acid metabolism, lipids are more effective fuels for every mole of O_2_ consumed, but carbohydrates produce energy faster than lipids [101]. Exhaustion of glycogen store in muscle and liver is linked to fatigue, weakness, and inattention of mind [102]. Thus, most endurance training athletes have to consume a high-carbohydrate diet.

Traditionally, for an athlete, adequate protein intake and intake duration are essential. Indeed, a high protein supply for athletes is quite important to ensure sufficient levels of amino acids to repair muscle damage caused by exercise, moderately increase the use of protein as an energy source, and improve the quality of lean tissue mass [103]. Dairy, lean meat, eggs, and soy are good sources of dietary protein that stimulate the mononuclear phagocyte system effectively, which activates immunity to reduce oxidative stress [104].

Lipid is also a valuable fuel source for endurance athletes. In addition, lipid is an important source of essential fatty acids, and is involved in signaling and transport, providing insulation and protecting vital organs [105]. The intake of fat involves the transport of fat to muscle cells, the binding and transport of fat in the cytoplasm, the regulation of intramuscular triglyceride synthesis and decomposition, and the transport of fat to mitochondria [101]. From a scientific point of view, lipid restriction (< 20% of total energy) may increase the risk of deficiency in fat-soluble vitamins, carotenoids, and essential fatty acids such as conjugated linoleic acids (CLA) [106,107]. Thus, a moderate intake of lipids (20% to 35% of total energy) is important to ensure an adequate intake of essential nutrients [105].

In addition, micronutrients are also important for sport performance [108]. Regular and intense training may meet basic organization maintenance and repair requirements, but decrease gastrointestinal absorption, increase nutrient losses by feces, sweat, and urine, as well as their degradation rates [109].

### 3.4. Dietary Pattern in Chronic Diseases and Oxidative Stress

#### 3.4.1. Diet and Mitochondrial Dysfunction

Mitochondria play a fundamental role in regulatory mechanisms. In fact, any mitochondrial dysfunction that cannot be reversed by an optimal and strong response to stress, can lead to energy dysregulation, cancer, apoptosis, aging, degenerative disorders. A mild oxidative stress response, through a DRP1-dependent type of mitophagy, regulates the dynamics between mitochondria-selective autophagy and mitochondrial biogenesis and fusion [110]. Polyphenols might exert a fundamental role in this complex pathway.

It is well known that most plant polyphenols act as antioxidants. These compounds mainly activate mild oxidative stress and stimulate positive and beneficial responses from the cells. It is speculated that they may play a role in scavenging reactive oxygen or nitrogen, and they act as signal molecules in the cross-talk between the mitochondrial-endoplasmic reticulum and enzyme pathways involved in energy balance [10].

Diet can regulate mitochondrial dysfunction. A study showed that ketogenic diet therapy (DTs) determined a significant clinical improvement of seizures and cognitive function in patients with Lennox-Gastaut syndrome (LGS) with mitochondrial dysfunction, a typical refractory epilepsy [111]. Supplementing berberine (BBR) can reverse the mitochondrial dysfunction caused by high-fat diet (HFD) and skeletal muscle hyperglycemia and can improve insulin sensitivity in rodent models of insulin resistance, partly due to increased mitochondrial biosynthesis. The effect of BBR on mitochondrial function and biogenesis depends on AMPK activation mediated by SIRT1 [112]. Supplementing the n-3 polyunsaturated fatty acid docosahexaenoic acid (DHA) is beneficial for patients with heart failure. DHA supplements can increase the DHA content in mitochondrial phospholipids by binding to membrane phospholipids, reduce the viscosity of mitochondrial membranes, and accelerate the uptake of Ca^2+^, leading to a reduction in the development of left ventricular dysfunction [113].

#### 3.4.2. Antioxidant Diet and Oxidative Stress

Antioxidant treatment could ameliorate vascular alterations associated with pro-inflammatory pathologies such as hypertension and diabetes. Supplementation of a wine pomace product rich in polyphenols to rats attenuated oxidative stress associated with hypertension and type 1 diabetes [114,115]. Both blood pressure and glucose were reduced by the wine pomace, and oxidative damage was lowered.

Wine pomace reduced wall aortic thickness, cross-sectional area, and wall/lumen ratio, decreased ROS, and increased eNOS activation. The administration of wine pomace product to hypertensive or diabetic rats exhibited protected against endothelial dysfunction and vascular remodeling. Wine pomace attenuated hyperglycemia and hypertension through reduction of oxidative stress and inflammation associated with the polyphenols it contains [116]. Yang et al. showed that the structures of o-diphenol and mdi-phenol play an important role in scavenging free radicals. A larger conjugation system in a functional molecule is conducive to a higher scavenging rate of free radicals [117]. When the chemical shift of phenol hydrogen is low, anti-oxygenation ability is strong. The fruit wine exhibits a strong scavenging ability on free radicals. It can inhibit the damage of red blood cells caused by the ·OH radical. Similar to phenolic compounds, sulfur-containing peptides (i.e., glutathione) and amino acids (i.e., cysteine) also have radical scavenging capacity [118].

Interestingly, resveratrol, a stilbene that is known to be present in red wine, mobilizes copper in human lymphocytes, contributing to the oxidative breakage of DNA in leukemias [119]. Therefore, the activity of phytochemicals might initially involve the induction of reactive species and, due to the powerful scavenging systems and the mild stress induced by low doses of polyphenols, result in the final activation of the indicated survival mechanisms. Similarly, blueberry wine exerts an outstanding antioxidative bioactivity due to its richness in anthocyanins and other polyphenols that exert putative health benefits through a moderate wine consumption [120].

#### 3.4.3. Diet in Obesity

Hepatic fat accumulation is caused by the synergistic effect of hepatic lipid dysregulation and pro-inflammatory cytokines via oxidative stress signaling pathways [121]. A high-fat diet will increase mitochondrial H_2_O_2_ production and insulin resistance [122]. A mitochondria-targeted antioxidant can prevent insulin resistance induced by a high-fat diet [123]. Several studies have reported that rats fed with a high-fat diet can develop liver fibrosis, and a dietary supplementation rich in antioxidants can effectively prevent liver fibrosis [124]. High-calorie diets not only affect the level of oxidative stress, but also have an adverse impact on offspring’s health. Miranda et al. [125] indicated that maternal high-fat diets increase the levels of hepatic cannabinoid receptors, endocannabinoid system triglycerides, and metabolizing enzymes; meanwhile, they improve the level of oxidative stress markers in the liver of male and female rat offspring but decrease the activities of catalase, superoxide dismutase, and glutathione peroxidase.

Polyphenols are characterized by a large number of phenolic groups, including a wide range of secondary plant metabolites. Of these chemicals, resveratrol has been shown to reduce high-fat diet-induced augmented endoplasmic reticulum markers in hepatocytes, body weight gain, serum triacylglycerols, total cholesterol, and LDL–cholesterol [126]. Yogurt supplementation in rats fed a high-fat diet has been shown to reduce the levels of oxidative stress markers in the liver and plasma, alleviate glucose intolerance, collagen deposition, inflammatory cell infiltration, and hepatic fibrosis [121].

#### 3.4.4. Diet in Non-Alcoholic Steatohepatitis (NASH)

NASH is characterized by high levels of de novo lipogenesis, lipid oxidation, ROS, and inflammatory cytokines [127]. NASH is diet-driven and characterized by inflammation and alterations in liver adipose fatty acid transport. De novo lipogenesis induces lipid oxidation and the production of ROS and inflammatory cytokines [127]. The pathogenesis of NASH is mainly based on “multiple parallel hit models”. Excessive accumulation of triglycerides in the liver will damage the mitochondrial respiratory chain. Liver is more susceptible to injury due to increased ROS, impaired antioxidant enzymes, and upregulated proinflammatory cytokines [128]. The main dietary factor contributing to NASH is increased consumption of refined carbohydrates and saturated fats [129]. Excessive consumption of omega-6 polyunsaturated fatty acids (n-6 PUFAs) may also increase liver disease through inflammation and oxidative stress [130].

Hepatic diseases are also associated with the accumulation of cellular lesions caused by the deregulation of redox homeostasis, which may result from increased ROS, such as superoxide and H_2_O_2_, or decreased antioxidant defense enzymes SOD, GPX, and CAT [109]. This disordered system leads to cellular oxidative stress, with decreases total thiols and increases residues of carbonylated proteins and derivatives from lipid peroxidation such as 4-HNE [131].

Fruits, tea, and vegetables contain polyphenols that have been shown to be beneficial to relieve NASH. Apple pomace added in Western diets has been reported to ameliorate oleic acid and palmitic acid transport from adipose tissue to the liver, downregulate inflammatory genes in the liver, and finally attenuate NASH [132]. Acai berry has also been reported to have a hepatoprotective effect against NAFLD in vivo [133], which is partly mediated by modulating the oxidant/antioxidant balance and inflammatory factors related to disease progression through inhibition of ROS production. More recently, the main measures to prevent steatosis and dyslipidemia involve the supplementation of α-linolenic acid and long-chain n-3 polyunsaturated fatty acids that can increase insulin sensitivity and inhibit adipogenesis [134].

#### 3.4.5. Diet in Type 2 Diabetes Mellitus (T2DM)

T2DM is a multifactorial and polygenic disease caused by insulin resistance and impaired insulin secretion. The important factors in the development of type 2 diabetes are insulin resistance and pancreatic B cell dysfunction [30]. T2DM dyslipidemia is not only a quantitative abnormality of lipoprotein but also a qualitative and dynamic abnormality with atherosclerosis potential. Typical quantitative abnormalities are hypertriglyceridemia and high-density lipoprotein (HDL)-cholesterol reduction [135]. Qualitative abnormalities include an increase in very-low-density lipoprotein–cholesterol and triglycerides and glycosylation of apolipoproteins [136]. Insulin resistance is a major factor for dyslipidemia in T2DM. Recent evidence suggests that adipokines such as adiponectin or retinol-binding protein 4 (RBP4) may play a direct role. In addition, HDL dysfunction in T2DM patients causes a loss of its anti-atherosclerotic properties, such as antioxidant capacity or endothelial-dependent vasodilation [137].

In recent years, epidemiological studies have demonstrated that a high intake of red meat and animal protein is related to an increased risk of T2DM [138]. However, a number of studies indicate that plant and animal protein diets improve glycemic control and reduce HbA1c [139]. High-protein diets are considered a promising and beneficial strategy for diabetes prevention and treatment because such diets can ameliorate the blood lipid profile and decrease liver fat and blood pressure [139]. In fact, both animal- and plant-based diets can reduce the content of protein carbonyls and MDA by alleviating oxidative stress [140].

#### 3.4.6. Diet in Cardiovascular Disease (CVD)

The risks for CVD are oxidative stress and systemic inflammation. For decades, cholesterol and saturated fat have been considered major dietary factors contributing to an increased risk of cerebrovascular and atherosclerosis diseases [141]. The increase of ROS can directly lead to modifications of lipid, DNA, or protein molecules, thus modulating signal transmission in cells, involving, for example, mitogen-activated protein kinase and redox-sensitive transcription factors [21]. The formation of atherosclerotic lesions is mainly induced by ROS mediating the activation of macrophages, changes in lipid expression, and formation of oxidized lipid products [142].

The Mediterranean diet is more effective than a relatively low-fat diet in preventing the development of CVD [83]. Dietary antioxidants have been shown to prevent CVD by quenching ROS and thereby interfere with the oxidative processes [143]. For example, dietary supplementation of the citrus flavonoid naringenin in mice fed a high-cholesterol diet mediated oxidative stress by regulating ATF6 activity, improving the lipoprotein profile and the dyslipidemia, and reducing atherosclerotic lesions [144].

Daily supplementation of ascorbic acid for eight weeks exerted protective effects against atherosclerosis and systemic inflammation by reducing the conversion of macrophages to foam cells and the formation of advanced glycation end products [145].

### 3.5. Diet in Neurodegenerative Diseases

The dietary pattern has been associated with a range of psychiatric disorders and neurodegenerative diseases, including Alzheimer’s diseases. Obesity and a high-fat diet may increase the incidence of AD, and the spatial memory and hippocampal synaptic plasticity of obese animal models are impaired [91]. High-fat diets may promote the expression of proinflammatory adipokines (IL-6, IL-1β, and TNF-α) and chemotactic adipokines (monocyte chemoattractant protein-1) and increase reactive microgliosis and astrocytosis [146]. Depression is also associated with a low antioxidant defense or failure to repair oxidative damage. There are many pathways connecting the brain with the gut, including the immune system response, the enteric nervous system, and metabolic processes of gut microbes [147]. Dietary components may modify Alzheimer’s disease by modulating the intracellular signaling pathways outlined above.

Many studies have shown that cognitive impairment is associated with decreased antioxidant capacity [148]. In large population studies, ascorbic acid and vitamin E supplementation showed a synergistic effect in alleviating Alzheimer’s disease by regulating oxidative stress [149]. Plant extracts and polyunsaturated fatty acids, e.g., big-leaf mulberry, rutin, resveratrol, EPA, and DHA, have been shown to alleviate stress [150,151,152].

## 4. Conclusions

Oxidative stress is a status of imbalance between oxidants and antioxidants. We highlighted the associations of diet patterns with oxidative stress and several metabolic diseases. High-calorie diets are considered one of the main factors leading to excessive ROS production, inducing obesity, neurodegenerative diseases, and immune inflammation. Foods rich in polyunsaturated fatty acids, fiber, and polyphenols may reduce the risk of chronic diseases by regulating oxidative stress. Dietary antioxidants may protect cells from oxidative damage by neutralizing ROS. Future challenges include the identification of interventions for oxidative stress and of the molecular mechanisms activated by diets and physical exercise, which are potential preventive and therapeutic targets for treating chronic metabolic inflammation and neurogenerative diseases.

## Figures and Tables

**Figure 1 foods-10-01854-f001:**
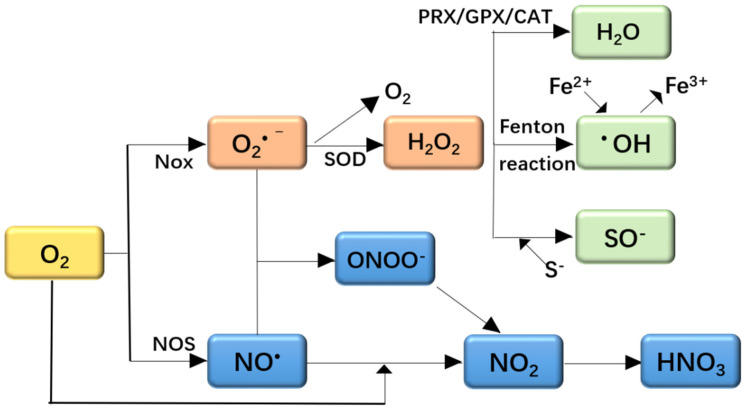
ROS and oxidative stress. NOX, NADPH oxidase family; ROS, reactive oxygen species; NOS, nitric oxide synthase; SOD, superoxide dismutase; PRX, peroxiredoxin; CAT, catalase; GPX, glutathione peroxidase; H_2_O_2_, hydrogen peroxide; ·OH, hydroxide radicals.

**Figure 2 foods-10-01854-f002:**
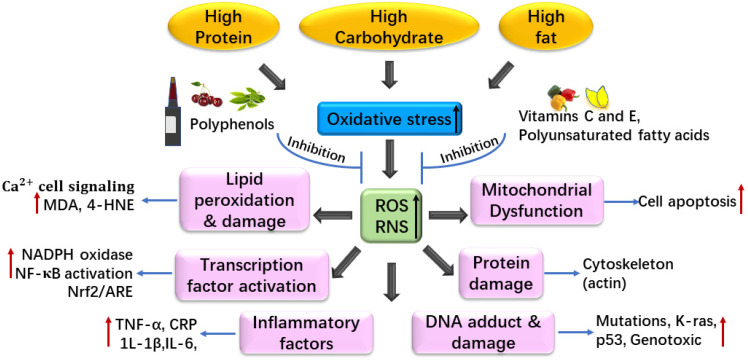
Associations between diet and oxidative stress. ROS, reactive oxygen species; RNS, reactive nitrogen species; MDA, malondialdehyde; 4-HNE, 4-hydroxynonenal; NADPH, triphosphopyridine nucleotide; TNF-α, tumor necrosis factor alpha; IL-6, interleukin-6; IL-1β, interleukin-1β; CRP, C-reactive protein; NF-κB, nuclear factor kappa-B; Nrf2, NF-E2-related factor 2; ARE, antioxidant response element; p53, a tumor suppressor gene; K-ras, a proto-oncogene.

**Figure 3 foods-10-01854-f003:**
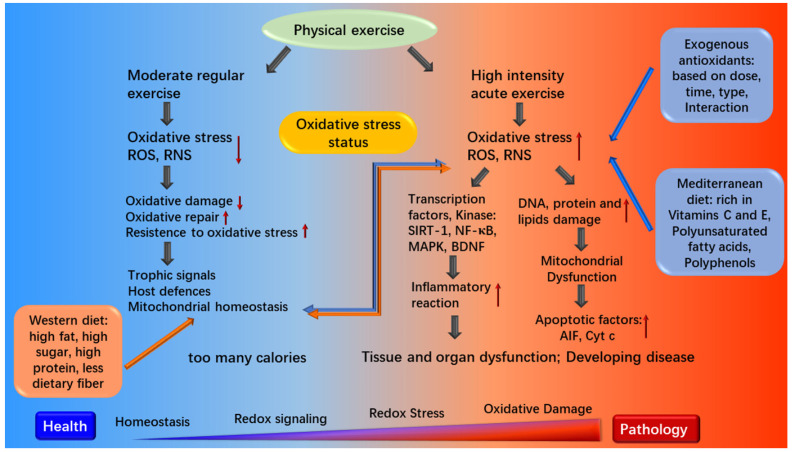
Associations between dietary patterns and physical exercise. SIRT-1, sirtuin1; MAPK, mitogen-activated protein kinase; BDNF, brain-derived neurotrophic factor; AIF, apoptosis inducing factor; Cyt c, Cytochrome C.

## Data Availability

Not applicable.

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
