# Peer review of "Dietary Regulation of Oxidative Stress in Chronic Metabolic Diseases"

_foods, 2021, doi:10.3390/foods10081854_

Round 1
Reviewer 1 Report
The manuscript resubmitted by Shuai Jiang et al. and entitled “Dietary regulation of oxidative stress in chronic metabolic diseases” is certainly an improved version of the original script. The investigators have addressed the critiques in satisfactory manner including a comprehensive literature search with added references. Although in the text there is a reference to Table 1, regarding the incidence of some chronic inflammatory diseases in Western diet population and Mediterranean diet population, the table is not included in this revised manuscript. Just make sure that it is not missing from the final version.
Overall, the revision was well-handled by the investigators and is worth considering publications.
Reviewer 2 Report
The Authors have modified the manuscript according to the suggestions. So, in my opinion, the manuscript can now be accepted.
This manuscript is a resubmission of an earlier submission. The following is a list of the peer review reports and author responses from that submission.
Round 1
Reviewer 1 Report
The manuscript entitled "Dietary regulation of oxidative stress in chronic metabolic diseases" was aimed at presenting an overview of the role of oxidative stress and diet in the development and progression of chronic disease. While this is an interesting and relevant area of research, several noticeable deficiencies considerably diminish the overall quality of this paper.
Major:
- Overall, the manuscript lacks structural organization, proper flow, and thus is difficult to follow.
- Overall structure and flow. The flow in the paragraphs can be improved by rearranging the information provided, thus that the sentences relate or build on each other. This can be achieved this by starting each sentence of a new paragraph by referring to information that was provided in the previous sentence before introducing new information.
- In the Introduction, the authors should clearly state the novelty and importance of this research, or how is this narrative review different from a similar review papers on the topic [e.g. Bjørklund G, Chirumbolo S. Role of oxidative stress and antioxidants in daily nutrition and human health. Nutrition. 2017 Jan;33:311-321].
- In its present form, the Discussion is rather convoluted and should be revised. It should better discuss and capture the major areas of agreement and disagreement in the literature on the given topic, provide its clinical significance, and provide the authors’ own interpretation of the presented findings. Current knowledge gaps should also be identified and rationales for future research provided. This would help highlight the importance of this work to the readers.
- Although, the Methods section is not required for narrative reviews, providing details on literature search strategies (e.g., databases, keywords, inclusion/exclusion criteria), is highly recommended as it allows the readers to reproduce the search results and/or determine potential for selection bias.
Minor:
- The manuscript has typos and grammatical errors (e.g., in the Title replace “dieases” with “diseases”) and would benefit from professional English language editing.
- Abbreviations should be defined at first mention and used consistently thereafter (e.g., Line 22 ROS).
Reviewer 2 Report
The manuscript by Shuai Jiang et al. entitled “Dietary regulation of oxidative stress in chronic metabolic diseases” is an interesting report on the development of dietary-driven oxidative stress related pathologies. The investigators have attempted to analyze the links between the common diets and metabolic imbalance that is due to excessive oxidative stress induced by certain foods. Oxidative stress has been indeed implicated in multiple chronic degenerative processes including those which affect the development of cancer and neurodegenerative disorders, atherosclerosis, inflammation, and aging, and even defense against infection. Diet seems to play an important role in development of some inflammatory conditions. The authors provided a comprehensive review of the molecular basis for the oxidative stress. However, as the effects of diet on metabolic homeostasis in humans are multifactorial and often overlapping, it is unclear which ingredient or number of such ingredients in diet is responsible for the final effect. Nevertheless, comparing the effects of Western diet on oxidative stress related pathologies with the Mediterranean diet is a reasonable approach. The analysis is clearly indicative so the impact of the report is important and may shed light on new avenues to maintain oxidative homeostasis by proper diet.
Having said that, the manuscript is of interest however more in-depth molecular mechanisms to control oxidative stress via oral administration (diet) should be presented. In fact, a discussion on recently published reports regarding systemic effects of orally administered foods would be of interest for this review. Here is a short list of references that are missing and worth further discussion in this interesting review:
- Schulze M. 2018. Food based dietary patterns and chronic disease prevention. BMJ 2018;361:j2396.
- Health effects of dietary risks in 195 countries, 1990–2017: a systematic analysis for the Global Burden of Disease Study. Lancet 2019; 393: 1958–72
- Kruzel M., et al. Insights into the Systemic Effects of Oral Lactoferrin: Transcriptome Profiling. Biochem Cell Biol. 2021 Feb;99(1):47-53.)
- Zimecki et al., The potential for Lactoferrin to reduce SARS-CoV-2 induced cytokine storm. Int Immunopharmacol . 2021 Mar 12;95:107571.
Here are my further questions/suggestions:
- The authors provide lots of common believes on regulation of many metabolic pathways by certain diets but the question is whether oxidative stress is indeed reduced by these foods? It is well known that oxidative stress induce mitochondrial dysfunction, but can diet control the mitochondrial dysfunction?
- The pathogenesis of chronic inflammatory diseases is not universal although the oxidative stress is a common damaging factor for most of them. High carbohydrate diet may alter the redox system imbalance but most likely has not effect on impairment of some body functions with already established inflammatory conditions such as obesity or AD. Question: Can diet revers the organ/individual pathology due to continuous oxidative stress? This is clearly missing from the manuscript and should be properly addressed.
- Is there a statistics for the Western diet population versus the Mediterranean one in frequency of AD or any other chronic inflammatory diseases? A table of such rate would be of value for this review manuscript. In fact a table for all diet-controlled pathological conditions would increase the visibility of this report.
- Associations between diet and oxidative stress Figure 2, the lower part (three bars) could be placed in a figure legend rather than being a part of scheme.
- Some of the acronyms/abbreviations need to be reviewed and corrected (e.g. ROS, RNS). Check for typos (“dieases” instead of disease in the title).
Overall the manuscript is well-written with no fundamental flows. However additional info is required to make this interesting report.
Reviewer 3 Report
Food 1195384 - Dietary regulation of oxidative stress in chronic metabolic dieases
There is a mistake in the title “dieases” instead of disease”.
OVERALL EVALUATION
In my opinion the work is very confusing, with several sentences truncated and disconnected from the context.
Furthermore, in many cases, for example see lines 323-329, Authors talk about the effect of molecules, in this case polyphenols, without explaining their mechanism of action.
I am not a native-English speaker, but the English style of the manuscript, need to be checked by a mother-tongue reviewer.
In my opinion the manuscript need a deep revision.
MAJOR REMARKS
Abstract. It is not clear to me want the Authors meant with “alleviate by proper exercise”. Did they mean “physical exercise “ ? If oxidative stress come from an unbalance between oxidants and antioxidants, I would have expected them to propose increasing the antioxidants intake or reduce the production of pro-oxidant molecules.
Page 2, lines 62-63. The oxidative stress has not “the capacity to remove them by the antioxidant systems”. Probably Authors meant “antioxidants defences”.
Page 4, line 167. What are “highly refined fats” ? In my opinion it is better say that Western diets are rich in “saturated fats”.
Page 6, lines 227-228. Fat and cholesterol absorption are not related to glycemic index. Fibers present in foods reduce glycemic index” but not “by preventing fat and cholesterol absorption” but “slowing down carbohydrate absorption”. Please correct the sentence.
Pages 6-7, lines 233-234. SCFA cannot be directly inserted into the diet, they are end-products of fiber fermentation. Therefore the sentence should be modified.
Page 7, line 247 and following. Wine is also a source of polyphenols, with anti-oxidants and vasodilating properties, these aspects need to be discussed.
Page 8, line 287. I disagree with the sentence. The respiratory quotient of fat is 0.7, while for carbohydrate is 1.0, in addition 1 mole of glucose produce 38 ATP while 1 mole of stearate 130 ATP. Carbohydrate produce energy faster than lipids, when substrate ATP is produced with anaerobic lactacid metabolism, but per mole of O2 consumed, lipids are mor efficient. The reference [89] of Spriet, deals with very peculiar situation and does not support a more efficient use of O2 by carbohydrate.
Page 8, line 288. “cholesterol store” or glycogen store ? This is an important mistake.
MINOR REMARKS
Page 1, line 10 and following. Please change “dietary mode” with “dietary habits” or “dietary pattern”.
Page 2, line 45. The acronyms ROS and RNS must be explained.
Page 2, line 56. Please use H2O2 instead of H2O2. See also line 85, 89 etc.
Page 5, line 174. “incidence of oxidative stress”, not “high incidence to oxidative stress”.
Page 7, line 236. Typing mistake: change “bining” in “binding”.
Page 9, line 311. In my opinion the word “between” is missed in action: “synergistic effect between hepatic lipid…”